# Understanding the Lost: Reconstruction of the Garden Design of Villa Peretti Montalto (Rome, Italy) for Urban Valorization

**DOI:** 10.3390/plants13010077

**Published:** 2023-12-26

**Authors:** Flavia Bartoli, Luca D’Amato, Arianna Nucera, Giulia Albani Rocchetti, Giulia Caneva

**Affiliations:** 1Institute of Heritage Science, National Research Council, ISPC-CNR, SP35d, 9, Montelibretti, 00010 Rome, Italy; flavia.bartoli@cnr.it; 2Department of Science, University of Roma Tre, Viale Marconi 446, 00146 Rome, Italy; giulia.albanirocchetti@uniroma3.it (G.A.R.); giulia.caneva@uniroma3.it (G.C.); 3Ministry of Culture, The National Gallery of Modern and Contemporary Art, 00197 Rome, Italy; arianna.nucera@cultura.gov.it; 4National Biodiversity Future Center (NBFC), University of Palermo, Piazza Marina 61, 90133 Palermo, Italy

**Keywords:** cultural heritage, floristic analysis, historical garden, historical landscape, landscape archaeology, lost gardens, urban regeneration

## Abstract

Urbanization and urban regeneration can significantly impact cultural heritage, but a greater knowledge of the past natural and historical features is needed to value the past and understand the present. The lost Villa Peretti Montalto in Rome, once located in the area that corresponds to the current front side of Termini station, deserves great attention due to its cultural value. This work aimed to provide a floristic and functional reconstruction of the gardens of the villa during the XVI and XVII Centuries. From several bibliographic and iconographic sources, a critical analysis and interpretation of plant names was conducted. A list of 87 species and their location in the different garden sectors, during different periods with their specific uses, is provided. The arboreal design was made by classical species in the Roman context, like *Cupressus sempervirens*, *Pinus pinea*, *Quercus ilex*, and *Ulmus glabra.* In addition, ancient lost varieties of fruit trees (mainly *Pyrus communis* and *Ficus carica*) and several species of conservation interest were found. The knowledge of the ancient flora in historical gardens could be a key tool in urban greenery planning and touristic and cultural valorization.

## 1. Introduction

Urbanization processes, including the land transformations associated with urban regeneration, can significantly impact the conservation of natural and cultural heritage [1,2,3]. The reorganization of the urban landscape reflects changing societal demands and adapts to land use, often resulting in significant impacts on culture and society [3,4,5]. Particularly in the historical cities of Europe that are rich in monuments and cultural heritage, the renewal or redevelopment of spaces and places over time has led to the loss of cultural landscapes, identity places, and, in many cases, historic buildings [1,6,7,8]. Such transformations have occurred widely, affecting gardens and their design, as highlighted by Ionescu et al., 2009 [6] and Cràciun, 2016 [9], who address the irreversible transformation of Bucharest (Romania) during the communist period.

Indeed, several examples of high-impact transformation can be found on all continents and under various regimes due to urban expansion, e.g., Maloney’s Garden in Sydney, Australia [10]; Stan Hywet Hall and Gardens in Akron, Ohio [11], New Jersey’s Stockton Gardens in the USA [12]; The Mughal Gardens and the Garden near Chauburji gate in Pakistan [13], and Kumasi, known as the “Garden city of West Africa”, in Ghana [14].

Tourism plays a fundamental role in urban regeneration, increasingly influencing urban planning agendas, with a growing focus on the conservation and valorization of cultural heritage [3,7,8]. Additionally, popular knowledge of the historical landscape strengthens the link between culture, environment, and community [6,15]. A case in point is Namwon City in Korea, which underwent a transitional process by shifting the hierarchical dominance in the urban structure from the Namwoneupseong Walled Town to the area of Gwanghalluwon Garden, influencing many transformation processes and nurturing a positive citizens’ perception of the Garden [16]. The transformation of a city over time is inevitable, but a greater understanding of past features is needed.

In this framework, the case of Rome is emblematic due to its three-millennium history and the richness of places and gardens with relevant cultural values [17]. Heavy transformations of the city also occurred in the modern age, as shown by the exceptional opportunity for a comparison spanning over four centuries of the different floras in the Colosseum, proving highly informative values of floristic data. This comparison revealed the rich plant colonization (684 species, mainly herbaceous ruderal plants, as the sum of the various lists) that appears to be due to the great variety of habitats in the monument and its state of abandonment. It allowed for a quantitative estimate of the decrease in the typical elements of mature dynamic stages, such as species related to grazing and animal droppings, weeds from cultivated fields, and, on the contrary, a constant increase in widely distributed and alien species [18,19,20,21]. Data also showed the climatic and microclimatic changes that occurred in the city were due to progressive warming and a reduction in humidity [22].

Further works showed that the transformation of ancient urban landscapes from the second half of the 19th century and just after the unification of Italy were related to the new projects for the new capital [23,24]. Many monumental gardens belonging to aristocratic families have disappeared due to heavy urbanization, such as Villa Ludovisi, Palombara, Giustiniani, and Astalli [25], which were famous for their high botanical and landscape values. An analysis of Nolli’s map of 1748 [26,27] also made it possible to quantify the ancient urbanistic structure of the city and the relevant role of vineyards or “*vigne*” (about 119 inside and 127 outside the ancient perimeter of the imperial Aurelian walls), villas and gardens, (99 inside and 28 outside), and 68 orchards (38 inside and 30 outside) [26,27].

Villa Peretti Montalto, owned by the powerful Cardinal Peretti, later named Pope Sixtus V, who initiated a major urbanistic transformation of the city of Rome, is an emblematic case of urban transformation [28]. Walking today in such a key area of Rome, it is not easy to imagine that until just a century and a half ago, it was occupied by a single, immense villa. Indeed, between 1559 and 1585, the now-called Termini area was affected by intense urban transformations, developed over different projects, contrasts, accelerations, and overlaps, marking the beginning of the process of converting eastern Rome into an essential area for building expansion [29].

In this study, we stress the need to include botanical information in the literature research when analyzing a past urban landscape. Based on the historic urban landscape (HUL) concept and the ICOMOS-Ename Charter, tangible and intangible aspects of the city’s image should be interpreted. Moreover, according to Sirisrisak and Akagawa (2007) [30], “vegetation” should be considered because it makes the historic urban landscape unique, and its interpretation and replacement history would help visitors understand not only the history of the place but also how local people interact with nature.

However, although archaeological sites often contain traces of ancient gardens and the study of ancient lost gardens has become a more popular discipline in recent decades [15,31,32], most studies analyze architectural, archaeological, or archaeobotanical data separately [15,33,34], and the related botanical information is scarce and fragmented. The conservation and valorization of a historical landscape and gardens must shed new light on the values accumulated over time and raise awareness of them [15].

Termini railway station, which is directly linked to the city and the international airport outside Rome, serves as the primary point of ingress to Italy for numerous visitors. As such, the present undertaking endeavors to unveil obscured historical information pertaining to this locale, aiming to rediscover erstwhile land utilization, encompassing antiquated cultivated species. This dataset stands poised as an invaluable instrument for illuminating the collective memory of a presently overlooked locale and augmenting the cultural significance of its natural attributes. This, in turn, holds the potential to be advantageous in crafting refined tourist itineraries.

To achieve this, we analyzed bibliographic, iconographic, and archivist documents and performed reconstruction work by comparing the ancient with the modern sources to find a match and recognize the garden’s floristic composition and each plant’s location and role. Moreover, we wish to provide further knowledge of plant taxa and their recurrences, such as uses and provenances, considering the general lack of a floristic reconstruction in the lost garden analysis, which is often approached by only considering artistic and architectural features.

## 2. Materials and Methods

The historical research initially concentrated on locating documents essential for reconstructing the physiognomy and design of the garden of the lost Villa Peretti Montalto. Another crucial step involved interpreting floristic data found in written and iconographic sources. Following the methodological proposal by Hosseini and Caneva [15], the reconstruction was accomplished through the integration and comparison of various historical, historiographical, and scientific sources, systematically organizing and cross-referencing them.

### 2.1. Archivistic and Bibliographic Collections

For the reconstruction of the gardens, several sources were utilized. We initially examined the study by Matthias Quast [35], who conducted a substantial review of archival documents. As a literary source, we also considered the poem by Aurelio Orso, published in 1588 and dedicated to Pope Peretti [36]. Consulting this text allowed us to extract valuable information about the garden, its components, and related floristic aspects. An additional source under consideration was the monograph published in 1836 by Vittorio Massimo [37], a member of the family that succeeded the Peretti Montalto dynasty in the ownership of the villa and witnessed its destruction. This work is of paramount importance as it compiles extensive information about the Termini area and the villa from antiquity up to the 19th century. The most comprehensive source of information regarding seventeenth-century transformations is represented by the “payment orders” of Cardinal Montalto, analyzed by Carla Benocci [38,39].

In addition to the aforementioned sources, two more documents dating back to that time yielded little but valuable information, which was interpreted and used for reconstruction: (i) the botanical treatise by Giovan Battista Ferrari, published in 1638 [40]; and (ii) Pietro De Sebastiani’s guide to Rome from 1683 [41].

### 2.2. Iconographic Interpretation for Mapping the Design of the Garden

To improve the information from written sources, we also carried out a comparison among the various iconographic sources, including urban maps [42,43,44], plans, axonometric representations, and the extensive collection of engravings (especially by Matthaus Greuter [45]), paintings, and drawings [46]. The foundational cartographic layer employed was Rome’s map from 1748 by Giovan Battista Nolli [43], which provides dimensions, main lines, and perimeter details [24]. We also used the interpretation of such a map by Mattias Quast, 1991 [35], comparing data from written sources with iconographic sources and excluding seventeenth-century additions. Both works allowed for the development of a basic planimetric scheme, and we translated it into a graphic format, specifically a zenithal representation, using AutoCAD 2018 and Adobe Photoshop (CS6 version) software, into which we inserted the botanical elements deriving from the floristic reconstruction. Through the use of GIS cartography, it was possible to overlay the map of the current city with the reference historical cartography (Nolli Map, 1748), identifying the exact correspondence of the villa in contemporary Rome.

### 2.3. Floristic Data and Botanical Planning Interpretation

Through bibliographic and archival research, we selected sources that documented the plants, their placement, and their uses in the garden [35,36,38,39,41,47,48,49,50]. Botanical elements are often described synthetically and generically in historical sources, sometimes in poetic form using common names. To conduct the taxonomical and synonymic evaluation of the mentioned entities, we followed the methods previously employed by Dinelli et al. (1994) [51] for interpreting names in Panaroli’s Flora of the Colosseum in Rome, given the similarity in ages and locations. We then conducted a comprehensive review of the terminology used by botanists in their respective times, based particularly on Matthioli (1568), Ferrari (1633), Panaroli (1643) [40,51,52] (taxa listed according to pre-Linnaean taxonomy), Penzig (1974), and Dinelli et al. (1994) ([51,53]; Linnaean taxonomy). Taxonomy was reported following Pignatti et al., 2017 [54] and http://www.worldfloraonline.org (accessed on 10 November 2023). In cases of doubt about interpretation, we critically evaluated each instance, considering the context of the original bibliographic sources and ecological and historical compatibility. Moreover, we consulted additional historical botanical sources to better assess taxonomy [55,56,57,58]. Following the historical sources, for each identified plant, we located and reported its placement in the specific garden area and its use (e.g., shade and ornamental trees, arborescent and high shrubs (SOs); creepers and hedges (CHs); herbaceous ornamental plants (HOs); fruit trees (FTs) vegetables (Vs); officinal and aromatic plants (OAs)) and attributed the botanical features of Life Form and chorological type according to Flora d’Italia [54]. We computed Life Form and chorological spectra to identify the main ecological features of plants at the site.

## 3. Results

### 3.1. Villa Peretti Montalto: History and Architectural Transformation of Termini’s Areas

The construction of the villa began in 1576, even before Sixtus V was elected Pope, by Guglielmini, together with the Cappelletti and Zerla *vigne*.

The first construction phase was completed in 1585, concurrently with the papal election of Cardinal Felice Peretti Montalto, and the former garden was enlarged and transformed into a wide, representative papal residence. Sixtus V, now Pope, appointed his sister Camilla Peretti as the property owner in 1586, and she managed it until her death in 1605 [49]. The estate experienced a first period of splendor during Sixtus V’s pontificate and then a second one at the beginning of the following century. With the end of the Peretti family, the estate passed through different hands until it came under the ownership of the Massimo family, who witnessed its gradual destruction between 1860 and 1888. The catalyst for this process was the construction of the first railway connections between Rome and its surrounding areas (Frascati, Civitavecchia, and Ceprano), all converging toward Termini. Initially, the villa housed the first station within some of its buildings, but subsequent construction by Salvatore Bianchi and the development of the new Esquilino district forced its progressive dismantling [28,46].

Villa Montalto occupied a large trapezoidal area bordered by four roads (Figure 1). This overlap between urban routes and the perimeter of the villa corresponds to the pre-existing ownership boundaries in favor of Pope Sixtus V and his sister Camilla Peretti, who simultaneously expanded their villa by acquiring land and constructing perimeter walls [59,60].

During the XVI Century, the area between the Basilica of *Santa Maria Maggiore* and the *Terme di Diocleziano* was essentially rural, mainly occupied by vegetable gardens and vineyards with a few houses, some churches, and convent buildings. The most notable elements were the basilicas of *Santa Maria Maggiore* and *Santa Maria degli Angeli*, the ancient ruins of the baths, and the hill forming the *Altissimus Romae Locus*. This area between the *Viminale* and *Esquilino* hills was chosen for the construction of noble villas due to its elevated position, healthy air, and the availability of water sources.

The villa, enclosed by straight stretches, extended for about 45 hectares; one-third was used as a private garden, and the remaining two-thirds were rented out to farmers, resembling a modern agricultural enterprise. The two areas were separated by internal walls, their layout likely influenced by pre-existing roads and farms, explaining the trapezoidal shape of the garden [35].

The main public access to the garden was from the solemn *Porta Quirinalis* and *Palazzo Termini* (later destroyed and rebuilt between 1883 and 1887 as the now-called *Palazzo Massimo*, hosting the National Roman Museum), at the present-day junction of Via *del Viminale* and Via *delle Terme di Diocleziano*.

Upon Camilla Peretti’s death in 1605, the villa passed into the hands of her nephew, Prince Michele, who entrusted the management of the property to his brother Alessandro. Cardinal Montalto (alias Alessandro Peretti Damasceni) was the influential patron of the seventeenth-century garden, and the design and execution of the garden were entrusted to the architect Alberto Martini [39]. They completed and enriched what was left by Pope Sixtus V upon his death: a complex with a clear overall design that was well-defined in the main avenues, focal points, and buildings but remained unfinished in some of its parts and furnishings (e.g., statues, sculptures, fountains, vegetation). Additional work began in 1606 and concluded with Alessandro’s death in 1623. They gave the villa an appearance that could be defined as definitive; after this date, the owners continued to manage it without making substantial changes [23,38].

Much later, in the concluding phase, several initiatives were undertaken. Firstly, the construction of two straight roads (via *Pia* and via *Merulana*); simultaneously, there was the partial arrangement of the *Terme di Diocleziano* with the construction of *S. Maria degli Angeli* and the new granaries, both overlooking the so-called *Piazza delle Terme* or *Termini* [28]. Two stately gardens overlooked this in parallel: the ancient thermal structures [23,62] inside the large exedra (now lost) and the less-known *Vigna Panzani* [63].

### 3.2. The Gardens and Their Floristic Reconstruction

#### 3.2.1. The Gardens of XVI Century

From the analysis of written and iconographical sources, a reconstruction of the sixteenth-century garden could be made. Seven main areas can be distinguished: the hanging gardens (HGs) and the triangular gardens (TGs) that represented the formal garden; the *Vigna* (V); the citrus grove (Cg); the upper garden (UG); the *Barchetto* (B); the rented orchards (ROs); and the *Orti della Cavallerizza* (OC) (Figure 2).

The formal garden, making up about 7% of the entire garden, was characterized by a geometrical pattern with a single axis of symmetry on which spaces (*Palazzo Termini* and the related buildings) and plant spaces were built, connected, and delimitated by tree-lined avenues [64] (Figure 2). The only significant difference in height was used to construct a terrace formed by a small centrally planned building—the *Casino Felice*—with side wings transformed into the hanging gardens, which constituted an extension of the magnificent house. The triangular gardens were situated at one of the two main entrances to the villa, an open-air vestibule that led to the entire complex. They served as the *pars dominica* or *urbana*, a public space independent of the building, utilized as a place of leisure for the owners and their guests. In these compartments, delimited by avenues and defined by hedges, officinal plants were cultivated [35,36], and a plant labyrinth was established [35].

Leaving the *loggia dei Leoni* on the upper floor, a *cavea scenica*, in the shape of a semi-circular theatre adorned with vines (*Vitis vinifera*) was introduced to the *Vigna* area, representing the so-called *pars fructifera*, i.e., the productive area of the garden. This extensive area was divided into four large compartments for different plantations. Firstly, there was the vine, but fifteen types of fruit species, olives, and five types of vegetables, such as cereals, were also produced. Numerous bushy species were likely planted to delimit the different cultivated lots. A *Citrus* grove (*Giardino dei merangoli*) was settled close to the triangular garden but was completely separated by walls that surrounded it on all sides. Citrons (*Citrus medica* L.) and lemons (*Citrus limon* L.) were also planted on espaliers along perimetral walls and those built to manage differences in heights [36,37].

A pergola avenue, approximately two hundred meters long and parallel to the casino connected the *Vigna* to the so-called upper gardens [36,64]. Unfortunately, historical sources do not provide detailed descriptions of this area, but based on the interpretation of documents, it was likely also intended for grape production [37]. In fact, it is possible that the Pope decided to leave this area as it was, postponing its planning to a later time, except for the two avenues that intersect it, forming a cross. The pergola was a prevalent element in Renaissance gardens and was generally composed of a vaulted wooden structure resting on wall elements or marble columns. Unfortunately, we do not know with certainty the plant characteristics of these pergolas, but they were probably adorned with vines (*Vitis vinifera* L.), as suggested by the iconographic sources of the painting in the *Loggia Dei Leoni* [50].

The southern corner of the garden was occupied by a *loco serato* made for the use of a *Barchetto* (or *Barco*, at the origin of the modern word “park”, indicating a wild and fenced space for hunting), an area of almost two hectares of natural vegetation wholly enclosed by walls and used as a small hunting area [23,64].

Finally, in 1590, shortly before the death of the Pope, Camilla Peretti dedicated an area of about thirty hectares of the external *Vigna* to rented orchards for the farmers [49]. From this moment, it became possible to rent the *Vigna* outside the garden to farmers, who could access them from the *Portae Exquilinae* located to the south.

#### 3.2.2. The Gardens of the XVII Century

Under the ownership of Cardinal Alessandro Peretti Damasceni, the extension of the formal gardens increased considerably, reaching about a quarter of the total (Figure 3). In the triangular gardens, the labyrinth disappeared, and two new fountains were installed, one for each compartment [64] (Figure 4a). There is no information on the maintenance of the espaliers of fruit trees. At the same time, the presence of ornamental trees was vaguely mentioned in all the main avenues of the villa. There is no floristic information about the hanging gardens, but we have references to new fountains and sculptural furnishingfrom Pinaroli (1725) and Tempesti (1754) [47,48].

The upper garden was the area that underwent the most significant transformations; it was divided into many small compartments by the “New Large Avenue” and many other minor transversal ones. The geometric compartments were defined trough boxwood (*Buxus sempervirens* L.) hedges and were probably characterized by *broderie*, arabesque parterres made with inert materials, such as earth, sand, stones, or earthenware, which were prevalent in gardens of the time [49]. Close to the main entrance of the villa, in front of the *Porta Quirinalis*, is a green theatre, and suggestive semi-circular space was created using evergreen species shaped and arranged to form scenic backdrops, in front of which statues and sculptures could be placed [49]. Two small sections of the upper garden were also transformed into groves, in which an oval-shaped fishpond (*Il Peschierone*) had been located with a sculptural group representing Neptune and Triton, designed by Gian Lorenzo Bernini (now hosted at the Victoria and Albert Museum in London) [49,65].

Furthermore, three of the five compartments of the *Vigna* became more extensive wooded areas [64]. The sources also testify the presence of a *ragnaia*, i.e., a plant design formed by a series of rows of trees and shrubs placed close together to form narrow, shady paths, which was probably located in this area. This shape was used to position the so-called *ragne*, special nets in which small birds could be caught [49]. In addition to the groves, the *Vigna* was characterized by the presence of an area designated as vegetable gardens located in the northeast corner. They were separated by a new tree-lined avenue that connected the access portals to the rent orchard area.

Transformations also concerned the extensive portion of rented orchards. The *Altissimus Romae Locus,* portrayed in the view by G.B. Falda (Figure 4c), was featured with a series of cypresses arranged in a circle around a statue, called the “Seated Rome”, and it was enriched by pine trees (*Pinus pinea* L.) and boxwood (*Buxus sempervirens* L.) espaliers, giving the place the appearance of a green theatre, similar to the other high ground.

Finally, the area, previously attributed as *Orto da Basso* or *Giardino dei Merangoli*, was divided into two parts surrounded by walls. One of these was occupied by the so-called *Orti della Cavallerizza* communicating with the *Menagerie* of Lions [42], areas used as a prairie where animals can circulate freely. The idea of inserting menageries in which to house rare or ferocious beasts was quite popular in the gardens of that time, one of the many stratagems used to arouse wonder and amazement in visitors and beyond [66]. Probably, the citrus fruits of the pre-existing *Giardino dei Merangoli* were transferred to the area adjacent to *Orti della Cavallerizza*, as shown by Greuter [46] (Figure 4b).

#### 3.2.3. Floristic Reconstruction

The floristic reconstruction of the sixteenth-century garden was supported by many written sources, enabling the reconstruction of almost the entire garden, except for the area of the upper garden and the *Barchetto*. On the contrary, for the seventeenth-century garden, information is scarce and not very detailed. We know with certainty that between 1606 and 1623, Cardinal Alessandro Montalto ordered the planting of 2697 new plants whose names, however, are unknown, except for a few cases. Later, after Alessandro’s death and the end of the Peretti–Montalto family, the villa passed into the hands of various families, who took care of the management of what already existed without making substantial changes.

The floristic reconstruction is shown in Table 1, comprising approximately 87 identified species, with around 20% identified only at the genus level, organized alphabetically based on the updated scientific names.

In terms of structure, there is a notable prevalence of phanerophytes (46%), followed by hemicryptophytes (19%). Geophytes and Therophytes are equally represented at 11%, with chamaephytes accounting for 10%. Regarding chorology, Mediterranean species dominate at 39.2%, followed by Asiatic species at 18% and Adventitious species at 6%.

#### 3.2.4. Botanical Plan of the Garden

The architecture of the garden evolved over time in the composition of the various garden areas (Table 2). In the sixteenth century, over 50% of the species were distributed in the formal garden area, particularly in the triangular garden and the hanging garden, with approximately 27% in the *Vigna* area. In the seventeenth century, the percentage of species in the *Vigna* area increased to 30%. An interesting aspect pertains to the shifts in the composition of the different garden areas over the centuries (Table 2) and the uses of plants.

In the XVI Century, the triangular gardens mainly hosted vegetables (39%), officinal and aromatic plants (28.5%), and fruit trees (14%), whereas the hanging garden hosted herbaceous ornamental plants (63%) and, in Vigna, 58% of the plants were fruit trees. Instead, in the XVII Century, the upper garden was highly modified, and the main plants were fruit trees and arborescent and high shrubs (90%). The triangular garden had 66% and 33% of arborescent and high shrubs. In this way, in the *Vigna* areas, 60% were fruit trees and 23% were arborescent and high shrubs (Figure 5).

In general, to enhance the perspective views along the tree-lined avenues, there were imposing trees such as cypresses (*Cupressus sempervirens*), pines (*Pinus pinea*), laurels (*Laurus nobilis*), and myrtles (*Myrtus communis*) [47,48]. Especially in the XVII Century, these species were also used to enhance statues or other ornaments with a circular arrangement and edge the areas bordering the avenues. The so-called *Viale dei Celsi* had particular features, starting from the main entrance and arriving at the other end of the villa, close to the wall parallel to the current via *Marsala*, and the name suggests the presence of mulberry trees (*Morus nigra*), whose existence in the orchard is certain in the stretch overlooked by a silk workshop.

In the sixteenth-century villa (Figure 4a), only a tiny portion was designed as a formal garden, i.e., the triangular compartments in front of the *Casino Felice* and the hanging gardens adjacent to it. In the former, officinal and aromatic plants were prevalent (*Citrus medica*) along with ornamental plants arranged in espaliers (*Citrus limon*). At the same time, in the latter, flowers of 26 different species were present. The flowerbeds delimited by the avenues mainly featured officinal plants, and a swirling labyrinth of greenery was set up inside one of the triangular compartments. All the remaining compartments, except for the grove, were intended to produce fruit, vegetables, and cereals. Among these, it is possible to distinguish a citrus grove called the *Giardino dei Merangoli* [49], a large portion of which was intended for the cultivation of grapevines (*Vitis vinifera*), and a *pomarium* with 17 different species of fruit trees, including olive trees (*Olea euroapea*). Furthermore, the presence of two pear varieties no longer cultivated should be underlined: the so-called *crustumina* (*Pyrus communis* var. *crustumina*) pear from the ancient *Crustumerium*, a town north of Rome already famous in the imperial era for producing this variety, and the *volema* (*Pyrus communis* var. *volema*), both of which were mentioned by ancient authors, like Pliny and Columella. In Cardinal Montalto’s garden in the seventeenth century (Figure 4b), the areas designated as formal gardens significantly increased and expanded into the so-called upper garden.

## 4. Discussion

In the most recent international scientific literature, the disappearance of traditional cultural landscapes and the emergence of new landscapes has become a recurring topic [2]. Sometimes, the changes are limited, but in other cases, such as Villa Montalto, they lead to the complete disappearance of the ancient landscape [1,2,69]. Indeed, Villa Montalto became fully part of the process that transformed an area that was still highly rural at the end of the sixteenth century into a highly urbanized one. The ancient presence of Villa Montalto influenced the urbanization process; in fact, several roads that we still travel on today have a close relationship with the villa: some coincided with its perimeter, and others led directly to its main portals, constituting privileged access points [28,46].

Between 1559 and 1585, the Termini areas were affected by significant urban transformation projects that were developed with different projects contrasting acceleration and overlaps and marking the beginning of the process of converting eastern Rome into an essential area of building expansion [29]. It is interesting to note how the same reasons that led Pope Sixtus V to choose Termini as the area on which to build his garden were decisive, centuries later, for their destruction: the healthiness of the area, located in an elevated position; the not excessive distance from the consolidated city; and the abundant presence of water. The *Altissimus Romae Locus* (Figure 3c) was one of the most characterizing elements of the villa and was formed by the *Aggere Serviano,* an embankment that served to improve military defense, dating back to the 6th century BCE. Over the centuries, it was progressively covered by earth and debris and became the highest point in eastern Rome. The construction of the Termini station at the end of the nineteenth century involved its excavation, leading to the discovery of the walls dating back to the Servian era, which are partially visible today on the *Piazza dei Cinquecento*. Virtual reality could be used to reconstruct the ancient lost landscape and represent the subsequent steps of landscape transformation as a tourist attraction, as has been performed elsewhere [70,71,72]. The use of augmented reality could be an immersive experience that enhances the understanding of the historical and cultural value of the ancient landscape and a tool for the valorization and dissemination of the abovementioned topics [70,71,72].

The comprehensive record of the plant species and their features in urban gardens can be fundamental within a valorization approach, providing essential insights into their plant composition, ecological dynamics, and historical significance. The lack of a detailed floristic inventory can lead to the wrong selection of plants during the valorization process, which often follows a selection of plants just for their colors and shapes or for the ease of their management, disregarding the ancient values of natural elements and the possibility of explaining them to the visitors [15]. On the other hand, in addition to the natural features of the garden, it is necessary to interpret and address the values of the ancient culture to explain contemporary ones [15,73].

The methodological steps required for nomenclatural reconstruction have been highlighted by Dinelli, Vinci, and Caneva [51]. It is desirable to carry out several steps and draw from several sources for the most reliable reconstruction possible. In the same way, it is necessary to highlight the considerable difficulties of interpretation that can be encountered in botanical reconstruction, especially when referring to the pre-Linnean era, in which the authors could also use completely personal nomenclatures. Comparing different bibliographic sources, we found different nomenclatural attributions to cite the same species, such as the viburnum (*Viburnum tinus*), cited in Aurelio Orso as “*Lentagini*” and in Matthioli as “*virburno*”, the bergamot (*C. aurantium* var. *bergamia*) as “*bergamotte*” or “*pomi di Adamo*”, or even the Marsh limestone (*Caltha palustris*) as “*Farfarugio*”.

Some ambiguities in the identification of species are dictated by synthetic and generic citations, in which the same term was often used in bibliographic resources to indicate different species that were systematically distant from each other. In fact, we see that the term “*Viola*” in Aurelio Orso’s text [36] is used to describe different species. It was possible to distinguish a species by carrying out an interpretative analysis of the text, such as “[…] *minio viole tingatur* […]” and “[…] *violae, quibus ora Leonis Adda feri* […]”, to describe two different species that are systematically distant from each other: *Viola* sp. pl. and *Anthirrinum* sp. pl., respectively.

However, it should be noted that in some cases, the terminology is highly specific, and although the attribution may seem uncertain, several sources report the same wording. The identification of the blueberry from the quote by Aurelio Orso “*nigro Vaccinia*” [36] may be questionable given the ecological features of the species, as it is quite distant from the geographical and climatic position of Rome, as well as its difficult cultivation, especially in gardens. Nevertheless, most of the consulted sources [55,56,58] refer to the blueberry with the same nomenclature.

Rediscovering the past urban landscape could be an essential tool for the cultural enhancement of a city, and knowledge of the past flora in historical gardens can be exploited in planning greener urban areas by combining their historical and naturalistic values in the process [13,74]. In this study, the presence of ancient lost or neglected varieties of fruit trees in the *Vigna area* emerged, such as *Pyrus communis* var. *crustumina* and *P. communis* var. *volema,* described by Plinius and Virgilius. Moreover, Aurelio Orso (1588) described different ancient varieties of *Ficus carica* cited in an almost derivative way from Plinius (in Naturalis Historia) and Cato (in *de agri cultura*) who reported, respectively “[…] *fici mariscae*”, described by Plinius as follows: “*si quidem et Lydiae, quae sunt purpureae, et mamillanae similitudinem earum habent et callistruthiae farti sapore praestantiores, ficorum omnium frigidissimae*” and “*Ficos mariscas in loco cretoso et aperto serito*” (Naturalis Historia, book XV, chapter 22; [67]). Therefore, this study highlights the potentiality of historic gardens and the knowledge of historical landscapes for the rediscovery of old and even lost horticultural varieties of ethnobotanical importance and the valorization of the cultural value of nature.

The flora of urban and peri-urban gardens can also include species of current conservation interest, providing data useful for the assessment of species’ presence, fluctuation, and ecology over time. In this study, the presence of noteworthy, uncommon taxa in modern gardens, e.g., *Inula helenium* subsp. *helenium, Vaccinium myrtillus, Caltha palustris, Nardostachys jatamansi, Ambrosia maritima,* and *Lycium europaeum,* emerged.

Moreover, in urban areas it is known that plants provide multiple ecosystem services: they contribute to improving ambient quality, mitigating the negative impacts of human activities, beautifying the anthropic environment, and promoting place identity and cultural heritage dissemination [75,76,77].

However, plant–human coexistence requires precise and adequate management measures, which are often ignored in cities’ government and planning. Therefore, to rediscover the use of species such as dogwood (*Cornus mas*, *Laburnum anagyroides*), German medlar (*Mespilus germanica*), and rowan (*Sorbus domestica*) as ornamental plants that are typical of our natural environment but wholly neglected for the street trees planning [78,79], we must consider the selection process in the context of urban greening and their ecological requirements [75].

Finally, several suggestions can be put forward to enhance the historical and cultural significance of this intangible heritage. It is widely recognized that plants play a fundamental role in the heritage landscape, serving as conduits for the history of a location. They are not mere ornaments but rather integral components that, when interpreted appropriately, bear evidence of historical landscapes. As such, they can serve as valuable tools for cultural valorization [15,27].

## 5. Conclusions

This study, providing the first floristic and functional reconstruction of the gardens of Villa Peretti Montalto in Rome during the XVI and XVII Centuries, sheds light on forgotten information about the history of the Termini area. Furthermore, we contribute to the understanding of plant taxa, their uses, and provenances. This is significant given the general lack of a floristic reconstruction in the analysis of lost gardens, which is often approached solely considering their artistic and architectural features.

This study serves as a valuable tool for highlighting the memory of a now-forgotten place, poised for valorization in tourist itineraries and future city regeneration. Virtual reality and augmented reality could be immersive experiences that allow the rediscovery of historical and cultural knowledge through the reconstruction of the ancient lost landscape and the representation of the subsequent steps of landscape transformations.

In this current period of innovation and improving living conditions, knowledge of historical landscapes and an understanding of modifications over time become fundamental tools for making informed choices for our cities. Therefore, this study serves as a starting point for bringing to light a piece of Rome’s landscape history that was almost completely forgotten, which should somehow be remembered and preserved.

## Figures and Tables

**Figure 1 plants-13-00077-f001:**
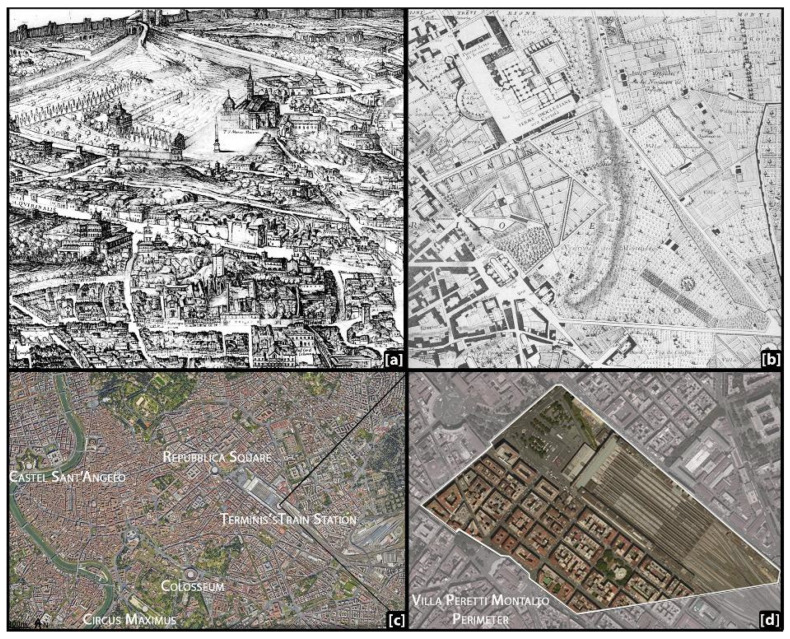
(**a**) A XVI Century map of Rome by Antonio Tempesta (1593, in Frutaz 1962 [44]) (**b**) at the time of Nolli, Rome, in 1748 (in Belli Barsali, 1963 [61]) and (**c**,**d**) today, with a highlighted detail of the perimeter of the lost Villa Peretti Montalto around Termini’s train station.

**Figure 2 plants-13-00077-f002:**
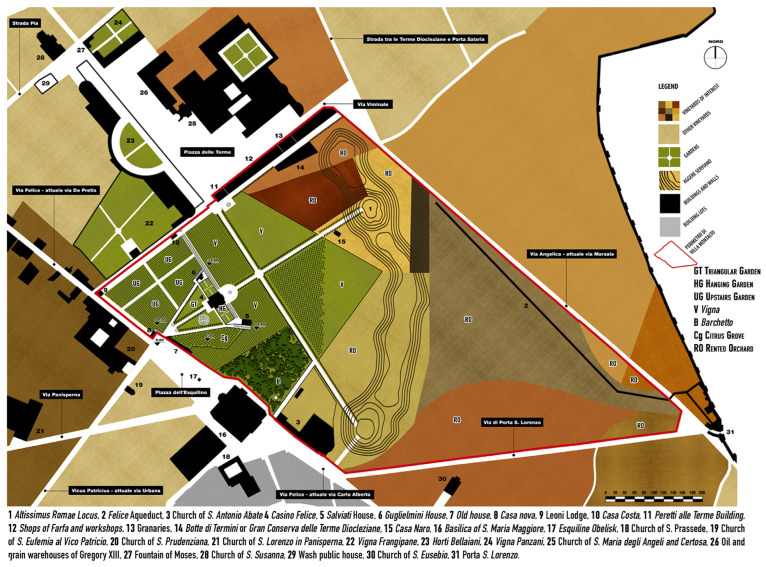
Reconstruction of the architectural transformations of Villa Peretti Montalto in the XVI Century.

**Figure 3 plants-13-00077-f003:**
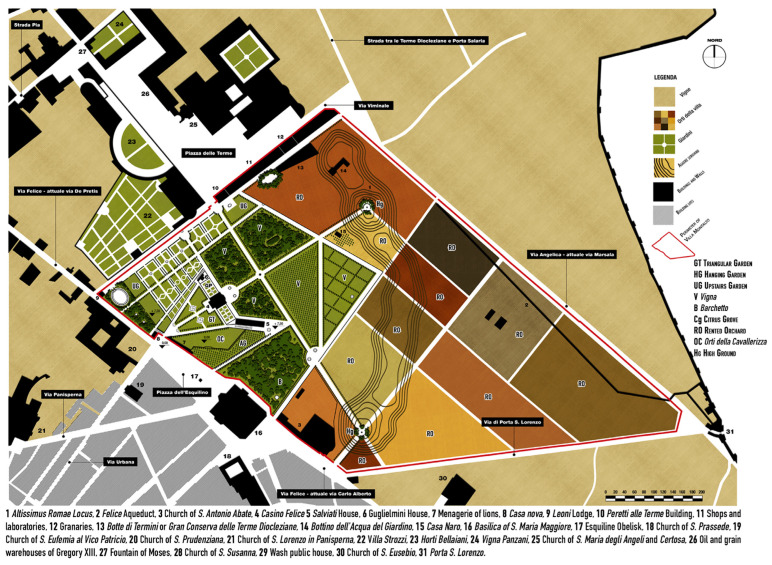
Reconstruction of the architectural transformations of Villa Peretti Montalto in the XVII Century.

**Figure 4 plants-13-00077-f004:**
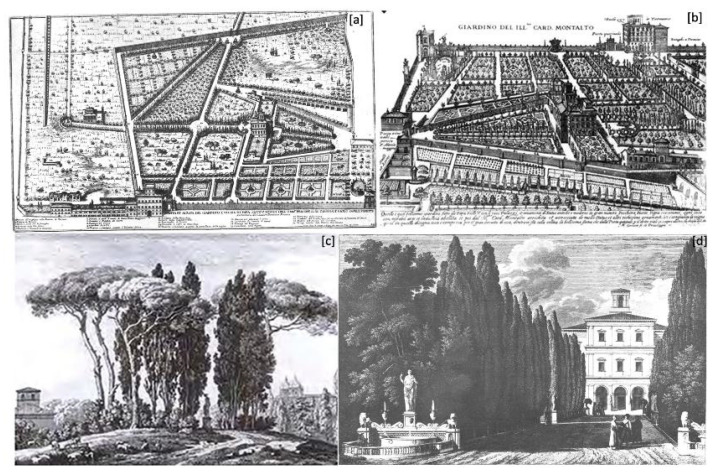
Historical views of the villa: (**a**) garden view of Villa Montalto (Incision of Giovan Battista Falda 1677); (**b**) view of the formal garden of the villa (Incision of Matthaus Greuter 1623); (**c**) view ofjustice hill (=*Altissimus Romae Locus*) (Bourgeois C. XVIII Century); (**d**) *Cupressus* tree line and view of casino felice (Percier and Fontaine).

**Figure 5 plants-13-00077-f005:**
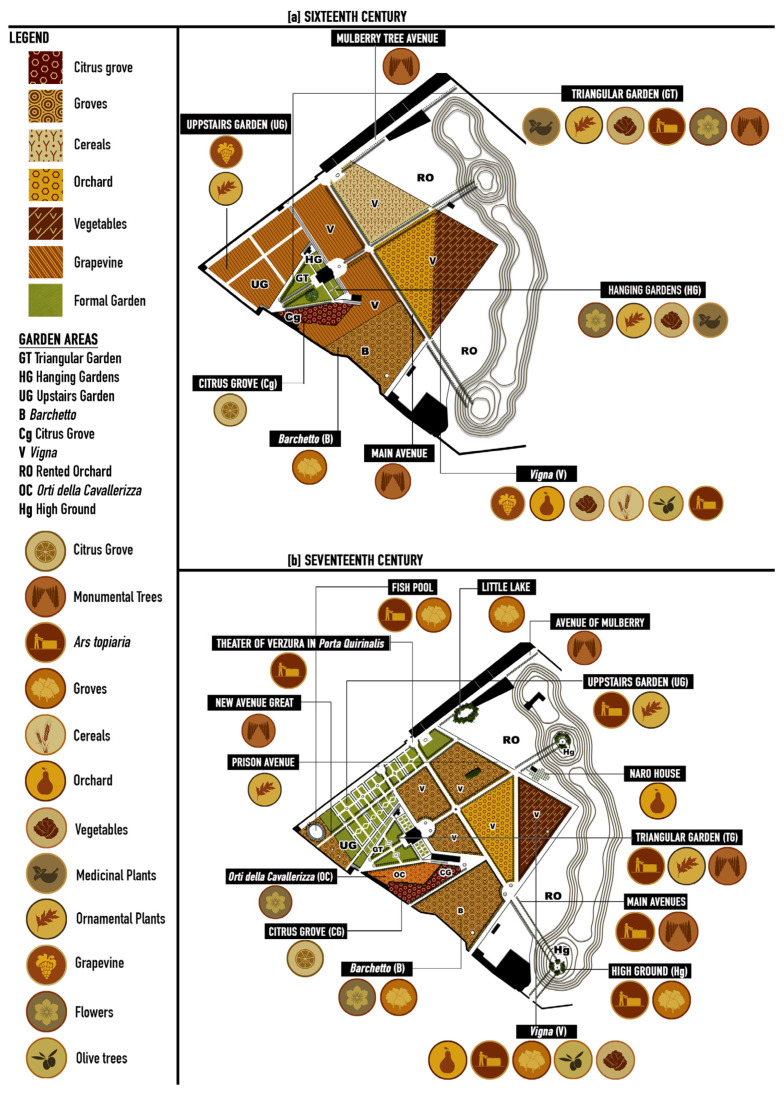
Garden structure and plant location in Villa Peretti Montalto in the sixteenth (**a**) and seventeenth (**b**) centuries.

**Table 1 plants-13-00077-t001:** Floristic list of plant species derived by synonymic evaluation of the entities mentioned in the ancient bibliographic and archival sources. The original names were reported in historical bibliographic sources (BSs): [A] Aurelio Orso, 1588 [36]; [B] Vittorio Massimo, 1836 [49]; [C] Matthias Quast, 1991 [35]; [D] Giovanni Pietro Pinaroli, 1725 [47]; [E] Guglielmo Blanco, 1618 [50]; [F] Carla Benocci, 1996 [38]; [G] Casimiro Tempesti, 1754 [48]; [H] Pietro De Sebastiani, 1683 [41]. The terminology was revised by botanists Matthioli (1568), Ferrari (1633), Panaroli (1643), Penzig (1974), and Dinelli, Vinci, and Caneva (1995) [51,52,53]. Accepted taxa of the plant species followed Pignatti et al., 2017; http://www.worldfloraonline.org (accessed on 10 November 2023), with their Life Forms and Chorotypes.

Bibliographic Sources	Original Name	Matthioli 1568	Panaroli, 1643 (Dinelli et al., 1995)	Ferrari, 1633	O. Penzig, 1974	Accepted Taxon	Life Form	Chorotype
**Shade and ornamental trees; arborescent and high shrubs (SOs)**								
[A], [B]	Cyparissus	Cipresso			*Cupressus sempervirens* L.	*Cupressus sempervirens* L.	P scap	Euri-Medit.-Orient.
[B], [C], [D], [F]	Lauro	Lauro			*Laurus nobilis* L.	*Laurus nobilis* L.	H scap	Steno-Medit.
[B], [F]	Pini	Pino			*Pinus pinea* L.	*Pinus pinea* L.	P scap	Euri-Medit.
[H]	Platani	Platano			*Platanus orientalis* L.	*Platanus orientalis* L.	P scap	SE-Europ
[F]	Pioppi	Popolo			*Populus* sp.	*Populus* sp.	P scap	n/a
[F]	Licinj				*Quercus ilex* L.	*Quercus ilex* L.	P caesp	Steno-Medit.
[A]	Casias, Casiae	Cassia Solutiua			*Cassia* sp.pl.	*Cassia fistula* L.	P scap	SE-Asiat.
[B]	Olmi	Olmo			*Ulmus montana* L.	*Ulmus glabra* Huds.	P scap	Europ.-Caucas.
[B]	Lentagini				*Viburnum tinus* L.	*Viburnum tinus* L.	P caesp	Steno-Medit.-Occid.
**Creepers and hedges (CHs)**								
[B]	Bussi	Bosso			*Buxus sempervirens* L.	*Buxus sempervirens* L.	NP, P ceasp	SubAtl-SubMedit
[B]	Gelsomini di Catalogna	Gelsomino		*Gelsomino di Spagno o di Catalogna*		*Jasminum grandiflorum* L.	P caesp	W-Asiat
[A]	Lygustro	Ligustro	*Ligustrum* (*Ligustrum vulgare* L.)		*Ligustrum vulgare* L.	*Ligustrum vulgare* L.	P caesp	Eurasiat.
[A], [G]	Mirtus, Mirti	Mirto		*Mortella*	*Myrtus communis* L.	*Myrtus communis* L.	P caesp	Steno-Medit.
[A]	nigro Vaccinia				*Vaccinium myrtllus* L.	*Vaccinium myrtllus* L. *	Ch frut	Circumbor.
**Herbaceous ornamental plants (HOs)**								
[A]	Acanthum	Acantho	*Acanthus* (*Acanthus mollis* L.)		*Acanthus mollis* L.	*Acanthus mollis* L.	H scap	Steno-Medit.-Occid
[A]	Adoni	Adonide	*Adonis* (*Adonis annua* L. ssp. *annua*)	*Anemoni*	*Adonis aestivalis* L., *A. autumnalis* L.	*Adonis annua* L.	T scap/G. bulb	Euri-Medit.-Steno-Medit.-Sett.
[A]	Amaranthi	Amaranto			*Amaranthus*	*Amaranthus* sp. pl.	T scap	n/a
[A]	Violae “quibus ora Leonis Adda feri […]”				*Anthirrinum* sp. pl.	*Antirrhinum* sp. pl.	Ch frut	n/a
[A]	Caltha			*Calta palustre*	*Caltha palustris* L.	*Caltha palustris* L.	H ros	Circumbor.-Eurosiber.
[A]	Cyanum	Ciano			*Centaurea cyanus* L.	*Centaurea cyanus* L.	T scap	Steno-Medit.-Subcosmop.
[A]	Crocus	Croco		*Croco*	*Crocus sativus* L.	*Crocus sativus* L.	G bulb	W-Asiat
[A]	Hyacinthum	Hiacintho		*Iacinti*	*Hyacinthus orientalis* L.	*Hyacinthus orientalis* L.	G bulb	E-Medit.
[A]	Iris	Iride		*Iride*	*Iris* sp. pl.	*Iris* sp. pl.	G rhiz	n/a
[E]	Citisi	Citiso			*Cytisus laburnum* L.	*Laburnum anagyroides* Medik.	P caesp	S-Europ.
[A]	Lilia	Giglio		*Giglio*	*Lilium candidum* L.	*Lilium candidum* L.	G bulb	E-Medit.
[A]	Floremque Jovis	Lichnide	*Lychnis sylvestris*, *Lychinis sylvestris altera*, *Lychnis viscosa* (*Lychnis* L., *Silene* L.)	*Licnide*	*Agrostemma flos jovis* L.	*Lychinis flos-jovis* (L.) Desr.	H scap	Subendem.
[A]	Leucoiis	Leucoio bianco et porporeo	*Leuconion* (*Verbascum* L., *Matthiola* R. Br. *Erysimum* L., *Alyssum* L.)		*Matthiola incana* L.	*Matthiola incana* (L.) W.T.Aiton	Ch suffr	Steno-Medit.
[A]	Narcise	Narcisso		*Narcisi*	*Narcissus tazetta* L.	*Narcissus* cfr. *tazetta* L.	G bulb	Steno-Medit.
[A]	Nardo					*Nardostachys jatamansi* (D.Don) DC. ***	G riz	Asiat.
[A]	Rubefacta papavera	Papavero selvatico	*Papaver erraticum, P. erraticu alteru*, (*P. rhoeas* L.), *Papaver spumosum*		*Papaver rhoeas* L.	*Papaver rhoeas* L.	T scap	Euri-Medit.
[A], [B]	Rosa	Rosa	*Rosa canina* (*R. canina* L.)	*Rosa*	*Rosa* sp. pl.	*Rosa* sp. pl.	n/a	n/a
[A]	Tymbrae	Thimbra o Satureia			*Satureja hortensis* L.	*Satureja hortensis* L.	T scap	Euri-Medit.-W-Asiat
[A]	Smilax	Smilace	*Smilax aspera* (*S. aspera* L.)		*Smilax aspera* L.	*Smilax aspera* L.	G rhiz	Subtrop.-Paleosubtrop.
[A]	Violas, Violae	Viola	*Viola*		*Viola* sp. pl.	*Viola* sp. pl.		
**Fruit trees (FTs)**								
[D]	Bergamotte				*Citrus limonium* Risso	*Citrus aurantium* var. *bergamia* (Risso and Poit.)	P scap	Adventitious
[C]	Limoni, Limoncelli	Limone			*Citrus limonum* Risso	*Citrus limon* (L.) Osbeck	P scap	Adventitious
[C]	Cedri	Cedro			*Citrus medica* L.	*Citrus medica* L.	P scap	Adventitious
[A]	Hesperis, Mala				*Citrus* sp. pl.	*Citrus* sp. pl.	P scap	Adventitious
[C], [B], [D]	Merangoli				*Citrus aurantium* L.	*C. x aurantium* L.	P scap	Adventitious
[A]	Corna	Corniolo			*Cornus mas* L.	*Cornus mas* L.	P caesp	Pont-SEEurop.-Steno-Medit.
[A]	Coryli	Nocciuole			*Corylus avellana* L.	*Corylus avellana* L.	P caesp	Europ.-Europ.-Caucas.
[A]	Cydonia	Cotogno			*Cydonia vulgaris* L.	*Cydonia oblonga* Mill.	P scap	W-Asiat
[A]	Ficus	Fichi	*Ficus sylvestris* (*F. carica* L.)		*Ficus carica* L.	*Ficus carica* L. **	P scap	Medit.-Turan.
[A]	Lydia					*Ficus carica* L. **	P scap	Medit.-Turan.
[A]	Lybisca					*Ficus carica* L. **	P scap	Medit.-Turan.
[A]	Mariscae					*Ficus carica* L. **	P scap	Medit.-Turan.
[A]	Nucem	Noce			*Juglans regia* L.	*Juglans regia* L.	P scap	W-Asiat
[A]	Mala	Mele		*Mele*	*Pyrus malus* L.	*Malus domestica* Borkh	P scap	Eurasiat.
[A]	Mespila	Nespolo			*Mespilus germanica* L.	*Mespilus germanica* L.	P caesp, P scap	Europ.-Pontica
[A], [B]	Morisvè	Moro			*Morus nigra* L.	*Morus nigra* L.	P scap	W-Asiat
[A]	Oleam	Olivo	*Olea sylvestris* (*O. europaea* L. var. *sylvestris* Hoffmgg. et Link)		*Olea europea* L.	*Olea europaea* L.	P caesp, P scap	Steno-Medit.
[A]	Cerasis	Ciregie selvatiche		*Ciriegio*	*Prunus avium* L.	*Prunus avium* (L.) L.	P scap	Eurasiat.-Pontica
[A]	Pruna	Pruno o Susino			*Prunus domestica* L.	*Prunus domestica* L.	P caesp, P scap	SW-Asiat.
[A]	Amydola	Mandorle			*Amygdalus communis* L.	*Prunus dulcis* (Mill.) D. A. Webb	P scap	S-Medit.
[A]	Persica	Pesco		*Pesco*	*Persica vulgaris* Mill.	*Persica persica* (L.) Batsch	P caesp, P scap	E-Asiat.
[A]	Punica	Melagrano			*Punica granatum* L.	*Punica granatum* L.	P scap	W-Asiat
[A]	Volème	Pere			*Pyrus* sp.	*Pyrus communis* L. var. *volema*	P scap	Medit
[A]	Pyra Crustumia	Pere			*Pyrus* sp.	*Pyrus communis* L. var. *crustumina*	P scap	Medit
[A]	Sorba	Sorbo			*Sorbus domestica* L.	*Sorbus domestica* L.	Pscap	Euri-Medit.
[A]	Vitea, Uvis, Uva	Vite vinifera		*Vite*	*Vitis vinifera* L.	*Vitis vinifera* L.	P lian	Unknown Origin
**Vegetables (Ves)**								
[A]	Caepa	Cipolla			*Allium cepa* L.	*Allium cepa* L.	G bulb	W-Asiatica
[A]	Porrum	Porro	*Porrum sylvestre* (*Allium porrum* L.)		*Allium porrum* L.	*Allium porrum* L.	G bulb	Medit.
[A]	Allia	Aglio domestico			*Allium sativum* L.	*Allium sativum* L.	G bulb	Asiatica
[A]	Betae	Beta o Bietola			*Beta vulgaris* L.	*Beta vulgaris* var. *cicla* L.	H scap	Euri-Medit.
[A]	Intyba	Endivia o Cichorea			*Cichorium intybus* L.	*Cichorium intybus* L.	H scap	Cosmop.
[A]	Coriandra	Coriandro			*Coriandrum sativum* L.	*Coriandrum sativum* L.	T scap	SW-Medit.
[A]	Cucumis	Cocomeri			*Cucumis sativus* L.	*Cucumis sativus* L.	T scap	Asiat
[A]	Cucurbita	Zucche			*Cucurbita* sp. pl.	*Cucurbita* sp. pl.	T scap	
[A]	Fraga	Fragaria		*Fragole*	*Fragaria vesca* L.	*Fragaria vesca* L.	H rept	Cosmop.-Eurasiat.-Eurosiber.
[A]	Caerea	Cereali			*Hordeum* sp. pl./*Triticum* sp. pl.	*Hordeum* sp. pl./*Triticum* sp. pl.	n/a	n/a
[A]	Picrim	Lattuca			*Lactuca sativa* L.	*Lactuca sativa* L.	H bienn	Unknown Origin
[A]	Mentha	Mentha			*Mentha* sp.pl.	*Mentha* sp.pl.	n/a	n/a
[A]	Amaracus	Amaraco			*Origanum majorana* L.	*Origanum majorana* L.	H scap	Saharo-Sind.
[A], [B]	Roris	Rosmarino			*Rosmarinus officinalis* L.	*Rosmarinus officinalis* L.	NP	Steno-Medit.
[A]	Rumex				*Rumex* sp. pl.	*Rumex* cfr. *acetosa* L.	H scap	Subcosmop.
[A]	Salvia	Salvia			*Salvia officinalis* L.	*Salvia officinalis* L.	Ch suffr	Steno-Medit.
[A]	Serpilla	Serpillo			*Thymus serpyllum* L.	*Thymus pulegioides* L.	Ch rept, Ch suffr	Eurasiat., Europ.
[A]	Thymum	Thimo			*Thymus* sp.pl.	*Thymus* sp.pl.	n/a	n/a
**Officinal and aromatic plants (OAs)**								
[A]	Ambrosium	Ambrosia, Botri			*Ambrosia maritima* L.	*Ambrosia maritima* L.	T scap	Euri-Medit
[A]	Abrotonum	Abrotano			*Artemisia abrotanum* L.	*Artemisia abrotanum* L.	Ch frut	Origin Unknown
[A]	Absynthia	Assenzo	*Absinthium romanum* (*Artemisia absinthium* L., *A. arborescens* L.)		*Artemisia absinthium* L.	*Artemisia absinthium* L. PC	Ch suffr, H scap	E-Medit-Eurasiat-Subcosmop.
[A]	Inulae	Enola o Helenio			*Inula helenium* L.	*Inula helenium* L. subsp. *helenium*	H scap	Orof. SE-Europ.
[A]	Panaceae, Panaces	Panace Heracleo			*Heracleum sphondylium* L.	*Heracleum* cfr. *sphondylium* L.	H scap	Paleotemp.
[B]	Spico	Lavanda			*Lavandula spica* Cav.	*Lavandula angustifolia* Mill.	NP	Steno-Medit.-Steno-Medit.-Occid.
[A]	Rhamni	Ramno	*Rhamnus primus* (*Lycium europaeum* L.)		*Lycium europaeum* L.	*Lycium europaeum* L.	NP	Euri-Medit.
[A]	Malachen	Malva	*Malva* (*Malva* sp. pl.)		*Malva* sp. pl.	*Malva* sp. pl.	n/a	n/a
[A]	Marrubium	Marrobio			*Marrubium vulgare* L.	*Marrubium vulgare* L.	H scap	Cosmop.-Euri-Medit.-Sudsiber.
[A]	Nasturtia	Nasturtio			*Nasturtium officinale* L.	*Nasturtium officinale* W.T.Aiton	H scap	Cosmop.
[A]	Papavera somno	Papavero selvatico, P. domestico, P. cornuto, P. spumeo	*Papaver erraticum*, *P. erraticu alteru*, (*P. rhoeas* L.), *Papaver spumosum*		*Papaver* sp. pl.	*Papaver* cfr. *sumnifreum* L.	T scap	Euri-Medit.
[A]	Rutae	Ruta	*Ruta sylvestris* (*Ruta chalepensis* L. or R. *angustifolia* Pers.)		*Ruta graveolens* L.	*Ruta* cfr. *graveolens* L.	Ch suffr	S-Europ.-S-Siber
[A]	Parthenice	Parthenio			*Pyrethrum parthenium* L.	*Tanacetum parthenium* (L.) Sch.Bip.	H scap	Eurasiat.-Asiat

* Dodoens, Rembert, 1517–1585 (1616). “Remberti Dodonaei Mechliniensis Medici Caesarei Stirpivm Historiae Pemptades Sex Sive Libri XXX”, Ex Officina Plantiniana apud Balthasarem et Ioannem Moretos, Antverpiae, Belgium, pag.768. [56]; Gerarde J. (1597). The Herball or Generall Historie of Plantes:of Whortes, or whortle berries. CHAP. 69. Iohn Norton, London. [55]; Johan Wilhelm Weinmann 1734 ca., n°744, references S43446 [58]. ** Pliny the Elder- Naturalis Historia—Liber XV [67]; Cato, De re rustica, 8. *** Grilli Caiola, M., Guarrera, P.M., Travaglini, A. (2014). *Le piante nella Bibbia*. Gangemi Editore, Roma [68].

**Table 2 plants-13-00077-t002:** Placement of plant species in the different garden sectors at different times with their specific uses. The hanging gardens (HGs); the yriangular gardens (TGs); the *Vigna* (V); the *Citrus* grove (Cg); the upper garden (UG); the *Barchetto* (B); the rented orchards (ROs); and the *Orti della Cavallerizza* (OC). Shade and ornamental trees, arborescent and high shrubs (SOs); creepers and hedges (CHs); herbaceous ornamental plants (HOs); fruit trees (FTs); vegetables (Ves); and officinal and aromatic plants (OAs). Plant species with no available data (n/a). Species with no location cited in the sources are not reported but are included in Table 1.

Uses	Taxon	Location in the Garden	
		XVI Cent.	XVII Cent.
*SO*	*Cassia fistula* L.	HG, TG	n/a
	*Cupressus sempervirens* L.	TG, V	H, TG, Ugs, V
	*Laurus nobilis* L.	UGs	Ugs, V
	*Pinus pinea* L.	n/a	H, TG, Ugs, V
	*Platanus orientalis* L.	n/a	Ugs, V
	*Populus* sp.	n/a	Ugs, V
	*Quercus ilex* L.	n/a	B, V
	*Ulmus glabra* Huds.	n/a	Ugs
	*Viburnum tinus* L.	V	V
*CH*	*Buxus sempervirens* L.	TG, V	H, Ugs, V
	*Ligustrum vulgare* L.	HG	n/a
	*Myrtus communis* L.	TG	V
	*Vaccinium myrtillus* L.	HG	n/a
*HO*	*Acanthus mollis* L.	HG	n/a
	*Adonis annua* L.	HG	n/a
	*Amaranthus* sp. pl.	HG	n/a
	*Antirrhinum* sp. pl.	HG	n/a
	*Caltha palustris* L.	HG	n/a
	*Centaurea cyanus* L.	HG	n/a
	*Crocus sativus* L.	HG	n/a
	*Hyacinthus orientalis* L.	HG	n/a
	*Iris* sp. pl.	HG	n/a
	*Laburnum anagyroides* Medik.	HG	n/a
	*Lilium candidum* L.	HG	n/a
	*Lychnis flos-jovis* (L.) Desr.	HG	n/a
	*Matthiola incana* (L.) R.Br.	HG	n/a
	*Narcissus* cfr. *tazetta* L.	HG	n/a
	*Nardostachys jatamansi* (D.Don) DC.	HG	n/a
	*Papaver rhoeas* L.	HG	n/a
	*Rosa* sp. pl.	HG, V	V
	*Satureja hortensis* L.	TG	n/a
	*Smilax aspera* L.	HG	n/a
	*Viola* sp. pl.	HG	n/a
*FT*	*Citrus limon* (L.) Osbeck	Cg, TG	Cg, TG, Ugs
	*Citrus medica* L.	Cg, TG	Cg, TG, Ugs
	*Citrus* sp. pl.	Cg, HG, V	Cg, Ugs, V
	*Citrus x aurantium* L.	Cg	Cg, OC, RO, Ugs
	*Citrus x bergamia* (Risso and Poit.)	Cg	Cg, Ugs
	*Cornus mas* L.	V	V
	*Corylus avellana* L.	V	V
	*Cydonia oblonga* Mill.	TG, V	TG, V
	*Ficus carica* L.	V	V
	*Juglans regia* L.	V	V
	*Malus domestica* Borkh	V	V
	*Mespilus germanica* L.	V	V
	*Morus nigra* L.	V	V
	*Olea europaea* L.	V	V
	*Prunus avium* (L.) L.	V	V
	*Prunus cerasus* L.	V	V
	*Prunus domestica* L.	V	V
	*Prunus dulcis* (Mill.) D. A. Webb	V	V
	*Prunus persica* (L.) Batsch	V	V
	*Punica granatum* L.	TG	RO, TG
	*Pyrus communis* L. var. *crustumina*	V	V
	*Pyrus communis* L. var. *volema*	V	V
	*Sorbus domestica* L.	V	V
	*Vitis vinifera* L.	Ugs, V	n/a
*Ve*	*Allium cepa* L.	V	n/a
	*Allium porrum* L.	V	n/a
	*Allium sativum* L.	V	n/a
	*Beta vulgaris var. cicla* L.	TG	n/a
	*Cichorium intybus* L.	TG	n/a
	*Coriandrum sativum* L.	TG	n/a
	*Cucumis sativus* L.	V	n/a
	*Cucurbita* sp. pl.	V	n/a
	*Fragaria vesca* L.	TG	n/a
	*Hordeum* sp. pl./*Triticum* sp. pl.	V	n/a
	*Lactuca sativa* L.	HG	n/a
	*Mentha* sp.pl.	TG	n/a
	*Origanum majorana* L.	TG	n/a
	*Rosmarinus officinalis* L.	TG, V	V
	*Rumex* cfr. *acetosa* L.	TG	n/a
	*Salvia officinalis* L.	TG	n/a
	*Thymus serpyllum* L.	TG	n/a
	*Thymus* sp.pl.	TG	n/a
*OA*	*Ambrosia maritima* L.	HG	n/a
	*Artemisia abrotanum* L.	HG	n/a
	*Artemisia absinthium* L.	TG	n/a
	*Inula helenium* L. subsp. *helenium*	TG	n/a
	*Heracleum* cfr. *sphondylium* L.	HG, TG	n/a
	*Lavandula angustifolia* Mill.	V	V
	*Lycium europaeum* L.	TG	n/a
	*Malva* sp. pl.	HG	n/a
	*Marrubium vulgare* L.	TG	n/a
	*Nasturtium officinale* R.Br.	TG	n/a
	*Papaver* cfr. *sumnifreum* L.	TG	n/a
	*Ruta* cfr. *graveolens* L.	TG	n/a
	*Tanacetum parthenium* (L.) Sch.Bip.	HG	n/a

## Data Availability

All data supporting the findings of this study are available within the paper.

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
