# Peer review of "Understanding the Lost: Reconstruction of the Garden Design of Villa Peretti Montalto (Rome, Italy) for Urban Valorization"

_plants, 2023, doi:10.3390/plants13010077_

Round 1

Reviewer 1 Report

Comments and Suggestions for Authors

Although the paper overall is great, the comments below may be helpful...

L23 overall of Pyrus à mainly Pyrus… (“overall of” has unclear meaning)

L37 some cases à many cases [“some” seems like an understatement]

L57 This paragraph is fine as it is, but the author might like to see and consider mentioning the following paper, which provides beautiful insight into the deep and complex human history of Rome, before and after the establishment of Rome as a city; the history of plant introductions and uses in Rome naturally accompanies the human history, but the latter is now much better understood perhaps.

Antonio, M. L., Z. Gao, H. M. Moots and e. al. (2019). "Ancient Rome: A genetic crossroads of Europe and the Mediterranean." Science 366: 708-714.

L93 Regarding the Termini station area, it might be good to highlight the fact that the international airport outside Rome is now directly connected to the city by a train that takes travellers directly to Termini (since not very many years ago). The station is now the first point of contact (literally) with the ground in Italy, for many travellers arriving in Italy; this is the location where the first impressions of Italy begin to be formed in the minds of many visitors.

L98 the plant’s allocation (meaning unclear) à each plant’s role

L100 end, delete unnecessary “the”

L101 considering their artistic àdelete “their” (it is not clear in this sentence what “their” refers to; the word is not needed and its removal makes the sentence easier to understand).

L121 Delete “In any case”, the change theàThe

L122 delete “the” after “regarding” (second “the” in this sentence is not needed and interrupts the flow for readers).

L125 deducedàinterpreted (latter word links nicely to the next section of the paper)

L184 Figure 1. Separate all four parts of the figure black borders; make letters a-d larger and delete the brackets around them (if journal allows); in the figure legend, part (a), state that “the Villa Peretti Montalto is located in the upper left, at the edge of an open area with fields”. This is the first image that shows the Villa, so it should be identified to help readers follow the text, and understand other parts of Figure 1.

L223 the reconstruction à a reconstruction (avoid using a second “the” in the same sentence, when possible, for better flow)

L224 was obtained à “was possible” or “could be made”

L240 use italics for botanical name, Vitis vinifera

L261 Add brief definitions/translations of Barchetto or Barco for readers not familiar with these Italian terms? Does it mean a little boat, or boat? Was there a pond in the garden, for evening visits?

L276 Figure 2. This should be expanded to use two pages, one page for each part of the figure, so that the keys and labels become readable. Perhaps two legends for two figures.

L277-278… à “in (a) sixteenth century, and (b) seventeenth century” This rearrangement makes it easier to relate the legend to the figure.

L319 “the reconstruct of almost the entirety of the garden” à “reconstruction of almost the entire garden”

L321 delete “the” before “information” (definite particle is not needed here)

L363 add guidance for readers:  Table 2…..are not reported. à are not reported, but are included in Table 1 [or list separately if not reported in Table 1?]

RE: Quercus ilex, located in the Barchetto: did acorns serve as an attractant for wild boar, in this area used for hunting? Or were they used to feed domestic pigs for the estate? (Can anything be said about this?).

RE: Citrus and Punica fruit trees….were these of special commercial value, and therefore planted in the Rented Orchards (Can anything be said about this?).

L409 “..the main portals,… access”à “..the main portals,… access points” [add “points”].

L413 Romae Locus was à Romae Locus (Fig. 3c) [i.e. use the figure to help the reader!]

Note re “the abundant presence of water”, it might be nice to note that even today, a (?)local water source supplies a public tap that can be used inside the Termini station [this reviewer has enjoyed the water, though he does not know if the source is still local].

L414 which was à and was

L416 In the centuries was progressively à In the centuries it was….

L421  subsequent step à subsequent steps; …in other cases à elsewhere [optional shorter expression]

L430 to valorise à to explain or valorise [Use a more neutral tone or expression? Maybe we cannot assume that an ancient culture will always valorise a contemporary culture; it may more commonly help explain a contemporary culture]

L431 The steps à The methodological steps…; delete “methodologically” after “have been”

L450 critical à uncertain? difficult? (“critical” does not seem correct in this context; the meaning is not clear).

L456 for the cultural à for cultural

L457 and the knowledge à and knowledge; then delete “is a tool that”

L458 could à can; “in planning urban greener by considering” à “in planning greener urban areas by combining

L459 after “…values” add “in the process”

i.e simplify the sentence and avoiding repetition of “tool”

L469 and the valorisation à and valorisation

Author Response

REV1

Although the paper overall is great, the comments below may be helpful...

Authors: Thank you for the comments. The authors will be pleased to respond and modify the text based on the feedback from the reviewers.

L23 overall of Pyrus à mainly Pyrus… (“overall of” has unclear meaning) Thank you for the comments. We have made modifications according to the reviewers' suggestions.

L37 some cases à many cases [“some” seems like an understatement] Thank you for the comments. We have made modifications according to the reviewers' suggestions.

L57 This paragraph is fine as it is, but the author might like to see and consider mentioning the following paper, which provides beautiful insight into the deep and complex human history of Rome, before and after the establishment of Rome as a city; the history of plant introductions and uses in Rome naturally accompanies the human history, but the latter is now much better understood perhaps.

Antonio, M. L., Z. Gao, H. M. Moots and e. al. (2019). "Ancient Rome: A genetic crossroads of Europe and the Mediterranean." Science 366: 708-714.

Thank you for the comments. Thank you very much for your appreciation! The authors are glad to inform that the suggested article has been included in the bibliography. 

L93 Regarding the Termini station area, it might be good to highlight the fact that the international airport outside Rome is now directly connected to the city by a train that takes travellers directly to Termini (since not very many years ago). The station is now the first point of contact (literally) with the ground in Italy, for many travellers arriving in Italy; this is the location where the first impressions of Italy begin to be formed in the minds of many visitors.

Thank you so much for the interesting suggestion; we have incorporated this intriguing point into the text

L98 the plant’s allocation (meaning unclear) à each plant’s role

Thank you for the comments. We have made modifications according to the reviewers' suggestions.

L100 end, delete unnecessary “the”

Thank you for the comments. We have made modifications according to the reviewers' suggestions.

L101 considering their artistic àdelete “their” (it is not clear in this sentence what “their” refers to; the word is not needed and its removal makes the sentence easier to understand).

Thank you for the comments. We have made modifications according to the reviewers' suggestions.

L121 Delete “In any case”, the change theàThe

Thank you for the comments. We have made modifications according to the reviewers' suggestions.

L122 delete “the” after “regarding” (second “the” in this sentence is not needed and interrupts the flow for readers).

Thank you for the comments. We have made modifications according to the reviewers' suggestions.

L125 deducedàinterpreted (latter word links nicely to the next section of the paper)

Thank you for the comments. We have made modifications according to the reviewers' suggestions.

L184 Figure 1. Separate all four parts of the figure black borders; make letters a-d larger and delete the brackets around them (if journal allows); in the figure legend, part (a), state that “the Villa Peretti Montalto is located in the upper left, at the edge of an open area with fields”. This is the first image that shows the Villa, so it should be identified to help readers follow the text, and understand other parts of Figure 1.

 Thank you for the comments. We have made modifications according to the reviewers' suggestions, we are sorry but we conserve the black background under the letters because it is more readable

L223 the reconstruction à a reconstruction (avoid using a second “the” in the same sentence, when possible, for better flow)

Thank you for the comments. We have made modifications according to the reviewers' suggestions.

L224 was obtained à “was possible” or “could be made”

Thank you for the comments. We have made modifications according to the reviewers' suggestions.

L240 use italics for botanical name, Vitis vinifera

Thank you for the comments. We have made modifications according to the reviewers' suggestions.

L261 Add brief definitions/translations of Barchetto or Barco for readers not familiar with these Italian terms? Does it mean a little boat, or boat? Was there a pond in the garden, for evening visits?

Thank you for the comments. We have made modifications according to the reviewers' suggestions.

L276 Figure 2. This should be expanded to use two pages, one page for each part of the figure, so that the keys and labels become readable. Perhaps two legends for two figures.

Thank you for the comments. We have made modifications according to the reviewers' suggestions.

L277-278… à “in (a) sixteenth century, and (b) seventeenth century” This rearrangement makes it easier to relate the legend to the figure.

 Thank you for the comments. We have made modifications according to the reviewers' suggestions.

L319 “the reconstruct of almost the entirety of the garden” à “reconstruction of almost the entire garden”

Thank you for the comments. We have made modifications according to the reviewers' suggestions.

L321 delete “the” before “information” (definite particle is not needed here)

 Thank you for the comments. We have made modifications according to the reviewers' suggestions.

L363 add guidance for readers:  Table 2…..are not reported. à are not reported, but are included in Table 1 [or list separately if not reported in Table 1?]

  Thank you for the comments. We have made modifications according to the reviewers' suggestions.

RE: Quercus ilex, located in the Barchetto: did acorns serve as an attractant for wild boar, in this area used for hunting? Or were they used to feed domestic pigs for the estate? (Can anything be said about this?).

We have not information about this, but we retain that the presence of Quercus ilex was related to natural vegetation of the area, tendentially, the Barco areas are not interested by gardening plan, indeed for this area we have less botanical information. We added a sentence in the description of the Barchetto area.

RE: Citrus and Punica fruit trees….were these of special commercial value, and therefore planted in the Rented Orchards (Can anything be said about this?).

It is known the commercial value of these species, but in detailed for Rented Orchard of Villa Peretti Montalto it is not possible assume that the use of this plants by the private farmers was related to this aspect.

L409 “..the main portals,… access”à “..the main portals,… access points” [add “points”].

Thank you for the comments. We have made modifications according to the reviewers' suggestions.

L413 Romae Locus was à Romae Locus (Fig. 3c) [i.e. use the figure to help the reader!]

 Thank you for the comments. We have made modifications according to the reviewers' suggestions.

Note re “the abundant presence of water”, it might be nice to note that even today, a (?)local water source supplies a public tap that can be used inside the Termini station [this reviewer has enjoyed the water, though he does not know if the source is still local].

 Thank you for the comments. Unfortunately, we are not aware of the current presence of spring sources near the termini district, while there are public fountains whose water comes from other sources (e.g., Peschiera, Acqua marcia, Salone vergine, and more). Further insight on water supply in Rome can be found in [1] Purcell, N. (2013). Rome and the management of water: environment, culture and power. In Human Landscapes in Classical Antiquity (pp. 194-226). Routledge.; [2] Evans, H. B. (1997). Water distribution in ancient Rome: The evidence of Frontinus. University of Michigan Press.; [3] De Kleijn, G. (2021). The water supply of ancient Rome: city area, water, and population (Vol. 22). Brill.

L414 which was à and was

Thank you for the comments. We have made modifications according to the reviewers' suggestions.

L416 In the centuries was progressively à In the centuries it was….

Thank you for the comments. We have made modifications according to the reviewers' suggestions.

L421  subsequent step à subsequent steps; …in other cases à elsewhere [optional shorter expression]

Thank you for the comments. We have made modifications according to the reviewers' suggestions.

L430 to valorise à to explain or valorise [Use a more neutral tone or expression? Maybe we cannot assume that an ancient culture will always valorise a contemporary culture; it may more commonly help explain a contemporary culture]

Thank you for the comments. We have made modifications according to the reviewers' suggestions.

L431 The steps à The methodological steps…; delete “methodologically” after “have been”

 Thank you for the comments. We have made modifications according to the reviewers' suggestions.

L450 critical à uncertain? difficult? (“critical” does not seem correct in this context; the meaning is not clear).

 Thank you for the comments. We have made modifications according to the reviewers' suggestions.

L456 for the cultural à for cultural

 Thank you for the comments. We have made modifications according to the reviewers' suggestions.

L457 and the knowledge à and knowledge; then delete “is a tool that”

Thank you for the comments. We have made modifications according to the reviewers' suggestions.

L458 could à can; “in planning urban greener by considering” à “in planning greener urban areas by combining

Thank you for the comments. We have made modifications according to the reviewers' suggestions.

L459 after “…values” add “in the process”

 Thank you for the comments. We have made modifications according to the reviewers' suggestions.

i.e simplify the sentence and avoiding repetition of “tool”

 Thank you for the comments. We have made modifications according to the reviewers' suggestions.

L469 and the valorisation à and valorisation

Thank you for the comments. We have made modifications according to the reviewers' suggestions.

Reviewer 2 Report

Comments and Suggestions for Authors

The review concerned an article entitled:  Understanding the Lost: Reconstruction of the Garden Design 2 of Villa Peretti Montalto (Rome, Italy) for the 3 Urban Valorisation

The text discusses the impact of urbanization and urban regeneration on cultural heritage, with a particular focus on the lost Villa Peretti Montalto in Rome. The villa, once situated in the present front side of the Termini Station railway, is highlighted for its cultural significance. The study aims to reconstruct the gardens of the Villa during the XVI and XVII centuries, providing insight into the flora and functionality of the site.

In conclusion, the work demonstrates the importance of understanding the historical and natural features of cultural heritage sites impacted by urbanization. The detailed reconstruction of the Villa Peretti Montalto's gardens provides not only a glimpse into the past but also offers practical insights for contemporary urban planning, emphasizing the significance of preserving and leveraging historical knowledge in the face of urban development. Im not fluent in English but I see also some language mistakes.

Questions and remarks:

Introduction

1.       The text mentions the significance of the Villa Peretti Montalto in Rome and its historical transformations. How does the study propose to use botanical information to shed light on the history and cultural values of this particular site?

2.       The discussion on the Colosseum's flora changes over four centuries highlights climatic and microclimatic shifts. Could the text delve deeper into how these changes are correlated with the monument's use and the broader environmental conditions in Rome?

3.       Line 61-63. The authors state that plants influenced changes in microclimate and humidity. However, recent research also highlights that they have the ability to purify the air from various types of pollutants (https://doi.org/10.1016/j.ecolind.2023.110259, https://doi.org/10.3390/su14052973 This is particularly significant, especially in Rome, where the term "miasmic air" was first used in connection with atmospheric pollution.

4.       The work aims to provide forgotten information on the Termini area's history. How does the study suggest this information could be utilized for future city regeneration and tourist itineraries, and what benefits might it bring to the local community?

Methodology

1.       The methodology incorporates the use of Autocad and Photoshop software for creating a zenithal representation. What considerations were taken into account to ensure the accuracy and reliability of the graphic representation, especially when translating information from historical sources into a modern digital format?

2.       In the iconographic interpretation section, the text mentions the utilization of various visual sources, including urban maps, plans, engravings, paintings, and drawings. How did the researchers address potential discrepancies or variations in these visual representations from different periods, and what challenges did they encounter in creating a cohesive planimetric reconstruction from these diverse sources?

Results

1.       Figure 2. The description on the figure are to small. They should be bigger to read it mor comfortable.

Discussion

1.       How has the disappearance of traditional cultural landscapes, such as Villa Montalto, impacted the transformation of urban areas, and what factors contributed to the complete disappearance of ancient landscapes in some cases?

2.       In what ways can virtual reality be utilized to reconstruct and represent ancient landscapes, like the Aggere Serviano of Villa Montalto, and how might this contribute to tourism and cultural understanding?

Conclusions

Authors can add information how can the floristic and functional reconstruction of the gardens of Villa Peretti Montalto contribute to the valorization of the Termini area in Rome, particularly in terms of integrating forgotten historical information into tourist itineraries and guiding future city regeneration efforts?

Author Response

Questions and remarks: Rev2

Introduction

  1. Rev2 The text mentions the significance of the Villa Peretti Montalto in Rome and its historical transformations. How does the study propose to use botanical information to shed light on the history and cultural values of this particular site?

Authors: Thanks for the question. Botanical information is fundamental to enhance the history and culture of a particular historical site, we improved with a sentence the aims in the introduction section.

  1. The discussion on the Colosseum's flora changes over four centuries highlights climatic and microclimatic shifts. Could the text delve deeper into how these changes are correlated with the monument's use and the broader environmental conditions in Rome?

 Authors: Thanks for the question, we added more detail about this point.

  1. Line 61-63. The authors state that plants influenced changes in microclimate and humidity. However, recent research also highlights that they have the ability to purify the air from various types of pollutants (https://doi.org/10.1016/j.ecolind.2023.110259, https://doi.org/10.3390/su14052973 This is particularly significant, especially in Rome, where the term "miasmic air" was first used in connection with atmospheric pollution.

Authors: Thanks for the suggestion, we modified the sentence in the introduction, because it can be misinterpreted. We only highlighted that the analysis of vegetation in the Colosseum area showed the occurrence of climatic and microclimatic changes during the time of the city, linked to the different rate of urbanization and the changes of the landscape. In any case, we added the suggested references and information in the discussion section.

  1. The work aims to provide forgotten information on the Termini area's history. How does the study suggest this information could be utilized for future city regeneration and tourist itineraries, and what benefits might it bring to the local community?

 Authors: Thanks for the question. We added major detail in the aims paragraph of the introduction section.

Methodology

  1. The methodology incorporates the use of Autocad and Photoshop software for creating a zenithal representation. What considerations were taken into account to ensure the accuracy and reliability of the graphic representation, especially when translating information from historical sources into a modern digital format?

Authors: Thanks for the question. We added major detail information in the main text.

  1. In the iconographic interpretation section, the text mentions the utilization of various visual sources, including urban maps, plans, engravings, paintings, and drawings. How did the researchers address potential discrepancies or variations in these visual representations from different periods, and what challenges did they encounter in creating a cohesive planimetric reconstruction from these diverse sources?

Authors: Thanks for the question.  Discrepancies or variations was recognized only for the floristic reconstruction, not for the planimetric and building information. In any case, the authors, clarified the sentences in the main text.

Results 

  1. Figure 2. The description on the figure are to small. They should be bigger to read it mor comfortable.

Discussion

 How has the disappearance of traditional cultural landscapes, such as Villa Montalto, impacted the transformation of urban areas, and what factors contributed to the complete disappearance of ancient landscapes in some cases?

Authors: Thanks for the question.  the authors, sentences in the main text, to clarify this aspect.

 In what ways can virtual reality be utilized to reconstruct and represent ancient landscapes, like the Aggere Serviano of Villa Montalto, and how might this contribute to tourism and cultural understanding?

Authors: Thanks for the question.  the authors, sentences in the main text, to clarify this aspect.

Conclusions

Authors can add information how can the floristic and functional reconstruction of the gardens of Villa Peretti Montalto contribute to the valorization of the Termini area in Rome, particularly in terms of integrating forgotten historical information into tourist itineraries and guiding future city regeneration efforts?

Authors: Thanks for the question.  the authors, sentences in the main text.